# Beyond dissolution: Xerostomia rinses affect composition and structure of biomimetic dental mineral in vitro

**Mikayla M. Moynahan** [ID]*, **Stephanie L. Wong**, **Alix C. Deymier** *

Dept. of Biomedical Engineering, School of Dental Medicine, UConn Health, Farmington, CT, United States of America

☯ These authors contributed equally to this work.
* Deymier@uchc.edu

**Data Availability Statement:** All relevant data are within the manuscript and its Supporting Information files.

**Funding:** This work was supported by Professor Alix Deymier's Startup Funds. M.M.M was funded through the University of Connecticut School of

## Abstract

Xerostomia, known as dry mouth, is caused by decreased salivary flow. Treatment with lubricating oral rinses provides temporary relief of dry mouth discomfort; however, it remains unclear how their composition affects mineralized dental tissues. Therefore, the objective of this study was to analyze the effects of common components in xerostomia oral rinses on biomimetic apatite with varying carbonate contents. Carbonated apatite was synthesized and exposed to one of the following solutions for 72 hours at varying pHs: water-based, phosphorus-containing (PBS), mucin-like containing (MLC), or fluoride-containing (FC) solutions. Post-exposure results indicated that apatite mass decreased irrespective of pH and solution composition, while solution buffering was pH dependent. Raman and X-ray diffraction analysis showed that the addition of phosphorus, mucin-like molecules, and fluoride in solution decreases mineral carbonate levels and changed the lattice spacing and crystallinity of bioapatite, indicative of dissolution/recrystallization processes. The mineral recrystallized into a less-carbonated apatite in the PBS and MLC solutions, and into fluorapatite in FC. Tap water did not affect the apatite lattice structure suggesting formation of a labile carbonate surface layer on apatite. These results reveal that solution composition can have varied and complex effects on dental mineral beyond dissolution, which can have long term consequences on mineral solubility and mechanics. Therefore, clinicians should consider these factors when advising treatments for xerostomia patients.

## Introduction

Xerostomia, commonly known as dry mouth, is a symptom caused by decreased salivary flow [1] that affects up to 64.8% of people depending on the selected population [2–4]. It is caused by a multitude of etiologies including medications, cancer treatments, and systemic diseases [1]. Xerostomia patients often suffer discomfort, dysphagia, irritations in mucosa, and considerable increased risk of caries and demineralization of the dental tissues [1, 5–7]. This increased caries risk is associated with changes in salivary quantity and composition; including decreased salivary pH and buffering capacity [1, 8, 9]. These compositional changes impede

Dental Medicine Summer Research Fellowship Funds. S.L.W was funded through the University of Connecticut Graduate School Biomedical Sciences Program. The funders had no role in study design, data collection and analysis, decision to publish, or preparation of the manuscript.

**Competing interests:** The authors have declared that no competing interests exist.

saliva's ability to neutralize acidic metabolites made by oral bacteria [7, 10] and increase dissolution of enamel and dentin [1, 11]. The healthy equilibrium of the oral environment becomes compromised due to these changes in salivary flow rate, composition, and pH.

The goals of most xerostomia treatments are to (1) relieve the sensation of dry mouth and (2) protect dentition from demineralization. Oral rinses are currently among the most commonly recommended treatments for xerostomia and use a number of compositional tools to fulfill the treatment goals. These include the use of an aqueous base and/or mucins and large lubricating molecules to provide a comfortable mouth feel, as well as high phosphate levels and/or fluoride doping to minimize dissolution. Overall, some ingredients in these rinses have been shown to provide temporary symptom relief of dry mouth [12, 13]. Numerous studies have investigated the effects of such oral rinses on dental dissolution and have seen mixed results, with some enhancing demineralization while others promote remineralization [14–21]. The problem with these erosion studies is that they each quantify erosion with varied measurement techniques [14–21] and only examined a solitary outcome for erosion. As a result, the current available research fails to truly elucidate the effect of oral rinse composition on dissolution and ignores the effect of these treatments on the remaining tooth mineral composition and structure. In addition, basic science studies examining the effects of specific ions/molecules on mineral content fail to provide clear clinically-relevant relationships about current treatment options [22, 23].

In this study, we seek to elucidate the effect of clinically-relevant oral rinse composition on the dissolution, composition and structure of biomimetic dental apatites. We applied a multi-technique approach to understand the effects of four different clinically-relevant solutions containing: aqueous bases, high phosphate levels, mucin-like molecules, and fluoride. The combination of mass loss, pH, Raman spectroscopy, Inductively Coupled Plasma Optical Emission Spectrometry (ICP-OES), and X-ray diffraction (XRD) measurements allowed us to provide a comprehensive view of how these solutions modify dental tissues at a compositional and structural level. This research will increase scientific understanding of the consequences of oral rinses beyond dissolution by elucidating their effects on remaining tooth solubility and strength. Due to the increased caries risk seen in xerostomia patients, this research will also help with clinical decision making when selecting xerostomia treatment options.

## Methods

### Synthesis of carbonated hydroxyapatite

Biomimetic dental apatites were prepared via aqueous precipitation methods as previously summarized here [24, 25]. Calcium nitrate and sodium phosphate were simultaneously added to 250 mL of MilliQ water heated to 60˚C dropwise at a rate of 1 mL/min. Varying levels of sodium bicarbonate were added to the initial aqueous solution to create crystals containing ~6, 8, and 10% wt% carbonate, equivalent to levels seen in dentin. All chemicals were sourced from ACROS Organics™, Fair Lawn, NJ. The pH was maintained between 8.9–9.1 using sodium hydroxide (NaOH). Following titration, the solution was kept at constant temperature (60˚C) and pH (9) for two hours with constant stirring. The resulting crystals containing ~6, 8, or 10% wt% carbonate were filtered in a Gooch Vacuum Filter, rinsed with deionized water, and dried at 60˚C overnight.

### Preparation of solutions

To determine the effects of oral rinse composition on biomimetic dental mineral, four solutions were selected: (1) Tap water (aqueous base) (2) 1X phosphate buffered saline (PBS—aqueous base with additional phosphate), (3) Biotène Dry Mouth Oral Rinse® (mucin like

containing—MLC), and (4) ACT® Anticavity Total Fluoride Mouthwash (fluoride containing—FC). The tap water was sourced from the Farmington, CT water supply. PBS was diluted from 10X PBS (Fisher Scientific, Waltham, MA) in MilliQ water. Biotène and ACT oral rinses were purchased from a local pharmacy. Biotène and ACT represent some of the most commonly prescribed oral rinses for xerostomia. Although they have complex compositions, these 2 solutions were selected for the presence of mucin-like molecules such as hydroxyethyl cellulose in Biotène and the presence of high levels of fluoride in ACT (S1 Table), making them unique and clinically-relevant solutions.

The original solutions were measured for initial pH, which are referred to as unmodified solutions. About 100 mL of each solution was adjusted to pHs of 5.5, 7.4, or 8.0 via the addition of either NaOH or hydrochloric acid (HCl) to represent changes to the oral environment. Since the unmodified pH of tap water was 8.0, the data for the unmodified and pH 8 conditions for tap water have been combined, resulting in a sample size of 6.

## Exposure of biomimetic apatites to solutions

50 milligrams of either 5.7, 8.4, or 10.0 ± 0.6 wt% $CO_3^{2-}$ apatite powder (respectively denoted as low, medium, and high $CO_3^{2-}$) was placed into a 15 mL centrifuge tube with 10 mL of either tap water, PBS, MLC, or FC solutions at varying pHs. These conditions examined 3 main variables: solution type, solution pH, and mineral carbonate content, creating a parameter space composed of 45 separate conditions.

Powders were agitated in the solutions for 72 hours during which the solution pH was measured at 1, 3, 6, 8, 24, 48, and 72 hr. After 72 hours, crystals and remaining solute were separated and collected after filtering through a medium fine filter paper (FisherBrand™, Waltham, MA). The following outcomes were characterized for each powder per condition, before and after solution exposure: mass loss, the carbonate levels, the lattice spacing, crystal size, and microstrain. The change in pH of the solutions, as well as dissolved solutes were also measured.

## Determination of apatite powder carbonate content with Raman spectroscopy

The composition of the powders was analyzed using a WITec alpha300vR Raman microscope with a 785 nm laser. 10 spectra per powder were obtained with a 50x objective and an integration time of 32 acquisitions x 0.5 seconds. Using the Witec Project 5.1 software, background corrections and advanced fitting features were used to fit the 431, 590, 950, 960, 1045, and 1070 $\Delta cm^{-1}$ peak areas using a Gaussian function. The carbonate to phosphate ($CO_3^{2-}/PO_4^{3-}$) ratio was calculated from the 1070 $\Delta cm^{-1}$ $\nu_1$ apatitic carbonate vibration peak area divided by the sum of the 950 and 960 $\Delta cm^{-1}$ $\nu_1$ apatitic phosphate vibration peaks areas (1070/950+960). $CO_3^{2-}/PO_4^{3-}$ ratios were averaged across all spectra for each condition and compared to unexposed powders to determine percent change. $CO_3^{2-}/PO_4^{3-}$ ratios were compared to instrument specific standard curves to obtain values of wt% carbonate for the powders.

**Measuring lattice parameters and crystal size with XRD.** X-ray diffraction (XRD) analysis was performed on one powder sample from each of the 45 conditions on a Bruker D2 Phaser X-Ray Diffractometer (Bruker AXS, Germany) with Cu Kα radiation (λ = 1.5406) operating at 30kV and 10mA. The samples were scanned at an acquisition rate of 1.0 s/step with a step size of 0.02˚ from 20˚- 60˚ (2Θ). XRD patterns were compared with standards from the ICDD powder diffraction file PDF-2 and PDF-4 database to identify any changes in phase or the presence of secondary phases. (002), (004), and (310) peaks of apatite were fit using a PseudoVoigt function to measure peaks centers and integral breadth. a- and c-axis lattice

parameters were calculated from the peak centers using Bragg's law. c-axis crystal size and microstrain were calculated from the integral breadth using the Halder-Wagner method [26].

### Determination of solute composition via ICP-OES

Solution solutes were analyzed via Inductively Coupled Plasma Optical Emission Spectrometry (ICP-OES) to determine the amount of calcium, phosphorus, sodium, and potassium dissolved into the solution during exposure. One sample from each condition was tested in a Perkin Elmer Optima 7300 Dual View ICP-OES at the Center for Environmental Sciences and Engineering laboratory at UConn. Samples were diluted in a stepwise fashion using 1% nitric acid: 0.5% hydrochloric solution until concentrations of the elements of interest were within the calibration range. Interference check solutions were analyzed (ICS A and ICS A+B; High Purity Standards, Charleston, SC) with all sample runs to compensate for matrix effects which may interfere with sample analysis. Standard quality assurance procedures were employed, including analysis of duplicate samples, method blanks, post digestion spiked samples, and laboratory control samples. Instrument response was evaluated initially, every 10 samples, and at the end of an analytical run using a calibration verification standard and blank.

### Statistical analysis

Statistical significance was determined for all quantitative outcomes including: mass loss, pH change, change in $CO_3^{2-}/PO_4^{3-}$ ratio, d-spacing, crystal size, microstrain, and calcium, phosphorus, sodium, and potassium solution content. Multi-way Analysis of Variance (ANOVAs) was used considering the factors of pH (5.5, 7.4, 8.0 or unmodified), carbonate level (low, medium, high) and solution type (PBS, MLC rinse, FC rinse, or tap water). Multiple sets of one-way ANOVAs were then run for each factor with Tukey Pairwise comparisons to identify significance for each independent variable. Significance was established as $p < 0.05$. All statistical analyses were performed with MiniTab (MiniTab, LLC, State College, PA).

Pearson's correlation coefficients were used to determine the relationship between the measured parameters and controlled factors. Statistical significance of samples was denoted as $p < 0.05$.

## Results

These experiments examined 3 main variables: solution type, solution pH, and mineral carbonate content, creating a parameter space composed of 45 separate conditions. The following outcomes were characterized for each powder per condition, pre and post dissolution: mass loss, the carbonate levels, the lattice spacing, crystal size, and microstrain. The change in pH of the solutions with time, as well as the elements dissolved into the solute were also measured to determine the effect of solution composition on biomimetic dental mineral dissolution/ recrystallization.

### Effects of water-based solution

Preexposure tap water contained ~0.015 g/L of sodium (Na), ~0.0019 g/L of potassium (K), ~0.0035 g/L of calcium (Ca), and non-detectable phosphorus (P) levels. This makes water the least mineralized solution and the only one without P. Solution Ca (up to 0.08 g/L) and P (up to 0.04 g/L) increased with exposure irrespective of initial pH ($pH_i$) and initial $CO_3^{2-}$ content (Figs 1A and 2A).

Solutions pH increased after powder exposure in all conditions, causing an average pH change ($\Delta pH$) of 2.8, 1.3, and 1.2 pH units for $pH_i$ of 5.5, 7.4, and 8.0 respectively. A decrease

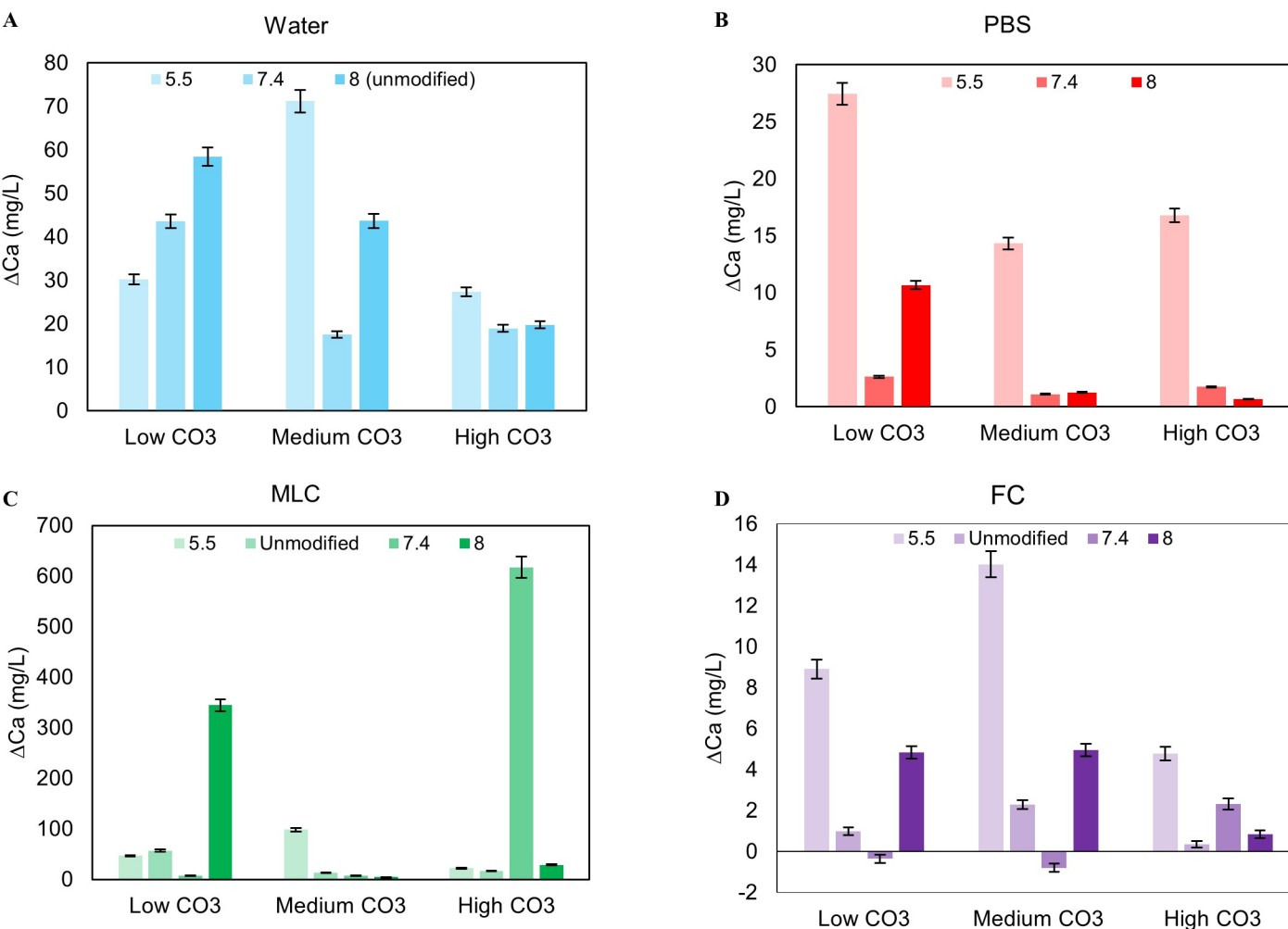

**Fig 1. Change in calcium levels in the solutions after exposure.** Calcium increased in all solutions, confirming apatite dissolution (**A-D**). Generally, calcium levels in the solution increased as $pH_i$ decreased for PBS (**B**). For water, MLC, and FC, there are no apparent trends (**A, C, D**). Overall, water had the most calcium in the solution after exposure since there is less calcium in water compared to the other solutions (**A**).

in $pH_i$ lead to an increase in $\Delta pH$ at all $CO_3^{2-}$ wt% (p = 0.000) while lower initial $CO_3^{2-}$ amounts led to a decrease in $\Delta pH$ (p = 0.000) at all $pH_i$'s (Fig 3A). The average mass loss in water was 0.011 g, with no statistically difference as a function of $pH_i$ or initial $CO_3^{2-}$ level (Fig 4A)

Water increased the $CO_3^{2-}/PO_4^{3-}$ of the apatite powders in all conditions (Fig 5A). No statistically significant relationship was identified between the change in $CO_3^{2-}/PO_4^{3-}$ and initial $CO_3^{2-}$ wt%, nor between $CO_3^{2-}/PO_4^{3-}$ and $pH_i$ in water. A higher final $CO_3^{2-}/PO_4^{3-}$ content weakly correlated to a decrease in Ca, P, and K ($c_{Ca}$ = -0.495; $c_P$ = -0.459; $c_K$ = -0.497) in the solution after exposure (Table 1).

Comparison of XRD spectra with standard spectra confirmed that the sample was hydroxyapatite or carbonated apatite with no secondary phases both before and after exposure. The average crystal size and average microstrain ($\varepsilon_{RMS}$) were variable irrespective of pH and initial $CO_3^{2-}$ content, suggesting that the structure is unaffected by water exposure (S1A Fig and Fig 6A). The (002) c-axis d-spacing ($d_C$) and the (310) a-axis d-spacing ($d_A$) showed no clear trends (Fig 7A). The $d_C$ decreased with increasing $\Delta CO_3^{2-}/PO_4^{3-}$ ($c_{002}$ = -0.413 and $c_{004}$ = -0.213) and decreasing final $CO_3^{2-}/PO_4^{3-}$ ($c_{002}$ = 0.500; $c_{004}$ = 0.388).

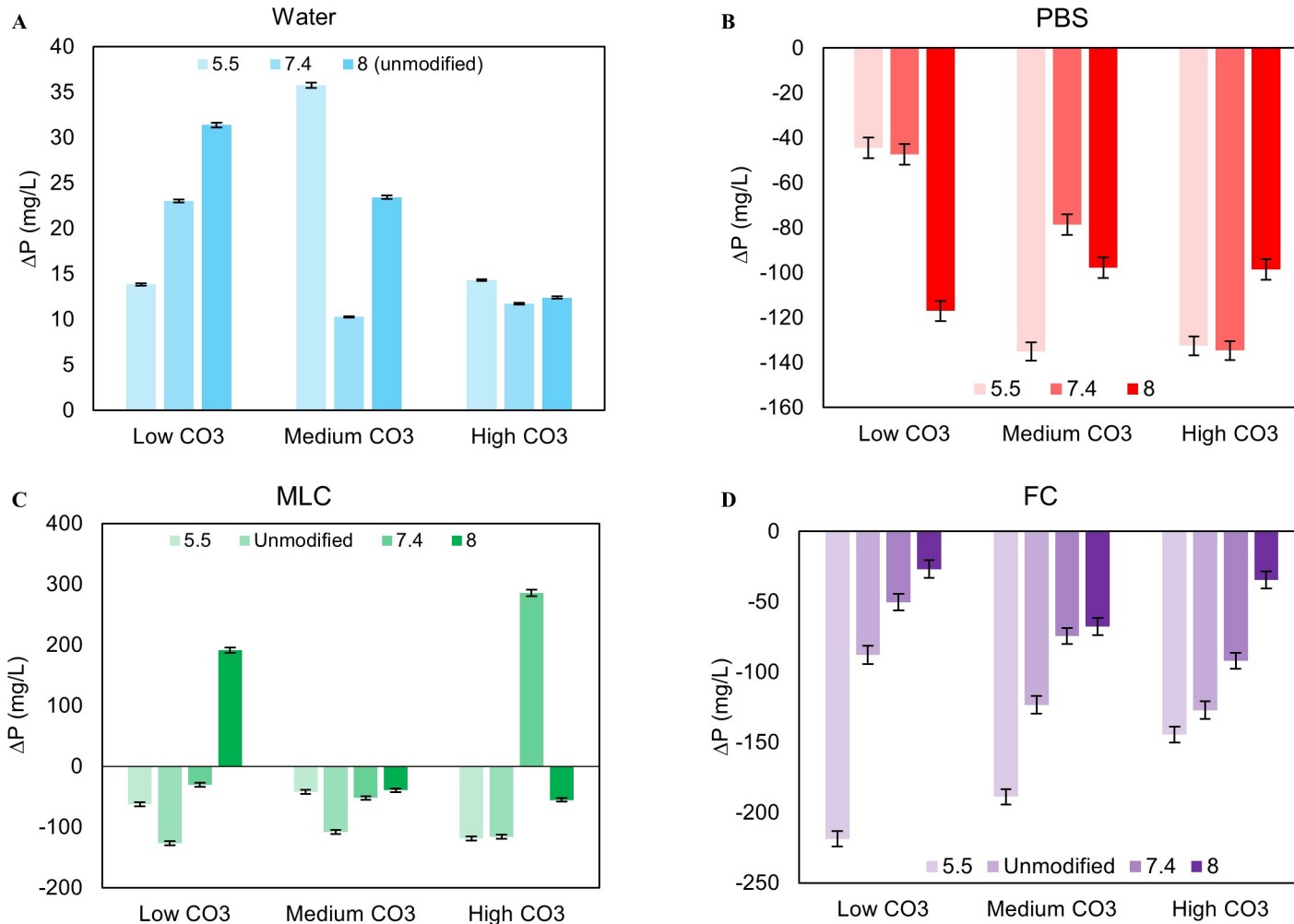

**Fig 2. Change in phosphorus amounts in the solution after exposure.** P decreased in all solutions except water, indicating an uptake of phosphorus in the apatite during recrystallization (**A-D**). There are no apparent trends in PBS and MLC (**B, C**). In FC, P decreased in the solution as initial $pH_i$ decreased (**D**). P increased in water due to the non-detectable levels of the initial solution (**A**).

### Effects of phosphorus-content in solution

Before dissolution at all pH's, PBS contained ~3.4 g/L of Na, ~0.4 g/L of P, ~$1.8 \cdot 10^{-7}$ g/L of K and ~ $4 \cdot 10^{-4}$ g/L of Ca. After exposure, Ca (up to 0.03 g/L) and K (up to 0.06 g/L) increased and P (down to -0.13 g/L) decreased (Figs 1B, 2B and S3B). The decrease in phosphorus is weakly correlated to $pH_i$ and $\Delta pH$ (correlation factors of 0.07 and -0.32, respectively, Table 1).

On average, pH increased by ~1, 0.4, and 0.5 pH units with decreasing initial pH. $\Delta pH$ was not affected by initial $CO_3^{2-}$, (Fig 3B) but decreased with increasing $pH_i$ (p = 0.000) (Fig 3B). $\Delta pH$ also increased with increasing Ca content (c = 0.597) (Table 1). PBS generated an average apatite mass loss of 0.011 g (Fig 4B), similar to water. These values showed no clear trends as a function of carbonate wt% or $pH_i$ of the solution.

Unlike the water-based solution with no phosphorus, powder carbonate levels decreased in PBS (Fig 5B). Higher initial $CO_3^{2-}$ levels in decreased $CO_3^{2-}$ content after exposure. At all $pH_i$, $CO_3^{2-}$ loss was reduced at low initial $CO_3^{2-}$ levels (p = 0.02) (Fig 5B). There was a negative correlation between $pH_i$ and $CO_3^{2-}$ loss (c = -0.677; p = 0.045) (Table 1). Final powder $CO_3^{2-}$ levels correlated negatively to the decrease in phosphorus in the solution (c = -0.520) (Table 1).

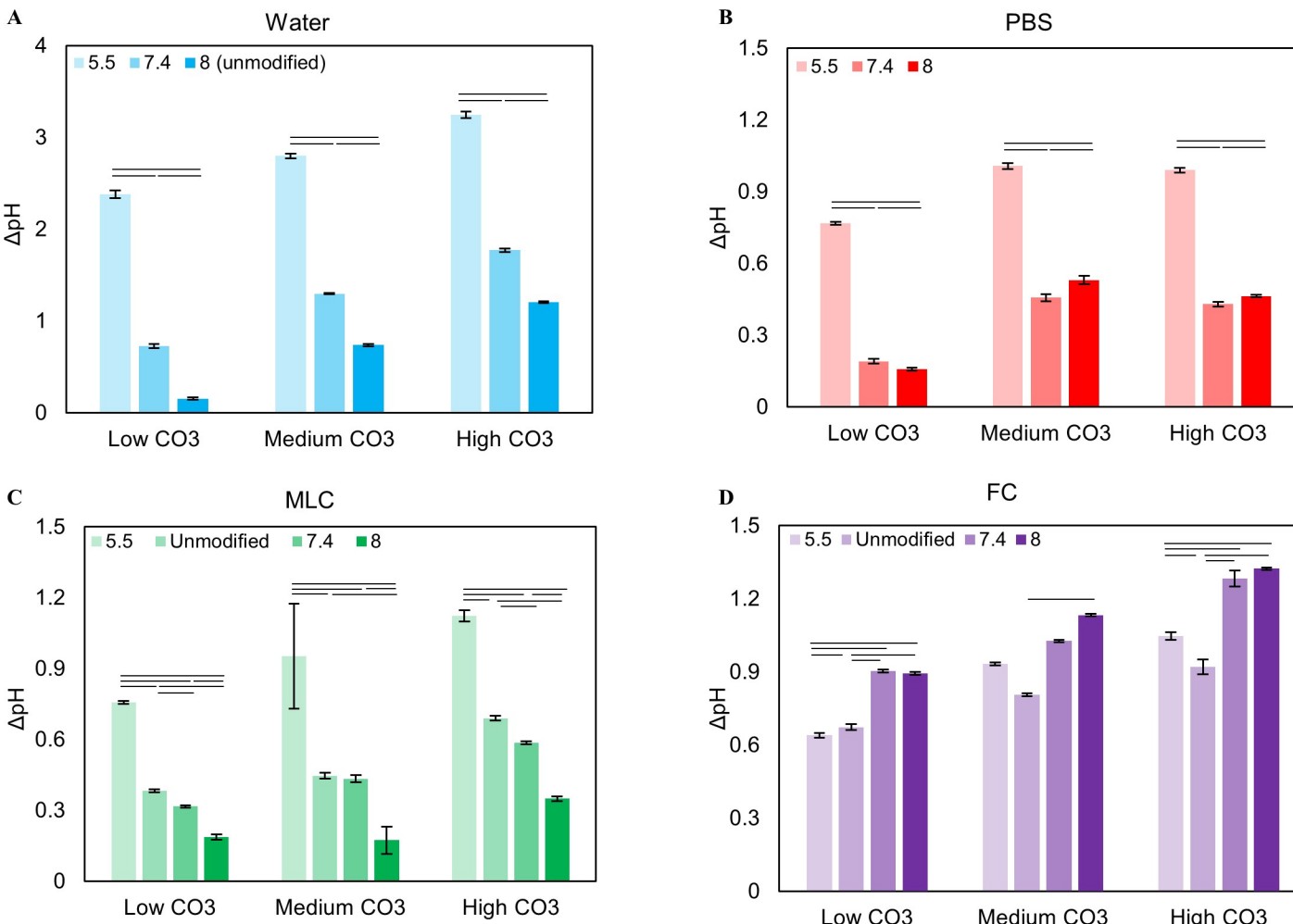

**Fig 3. The change in pH (ΔpH) of the solutions as a function of initial pH.** ΔpH increased as initial pH decreased for Water, PBS, and MLC, indicating that the powder is buffering the solution more at lower pHs **(A, B, C)**. For FC, ΔpH increased as initial pH increased **(D)**.

XRD results indicated that $d_C$ generally decreased while $d_A$ increased with exposure. The change in c-axis d-spacing ($\Delta d_C$) decreased as the $CO_3^{2-}$ content increased (Fig 7B). The change in a-axis d-spacing ($\Delta d_A$) did not change with initial $CO_3^{2-}$ content nor $pH_i$. PBS increased the average crystal size of the powders by 0.3–1.4 nm (S1B Fig). Overall, there were no clear trends in size change as a function of $pH_i$ or $CO_3^{2-}$ content. Conversely, the micro-strain ($\varepsilon_{RMS}$) on the crystals decreased after phosphorus exposure suggesting the presence of more uniform and organized crystals (Fig 6B). The $\varepsilon_{RMS}$ showed no clear trends with either $pH_i$ or $CO_3^{2-}$ content. Correlation studies found that an increase in final $CO_3^{2-}/PO_4^{3-}$ in the powders exposed to phosphorus correlates positively to a significant increase in $d_C$ ($c_{002}$ = 0.83; $c_{004}$ = 0.79) ($p \leq 0.01$) and negatively with $d_A$ ($c_{310}$ = -0.58) (Table 1).

## Effects of mucin-like molecules in solution

Pre-exposure MLC solution contained ~1.3 g/L Na, ~0.36 g/L P, ~0.015 g/L K, and ~0.0006–0.0014 g/L Ca for all pH's. Post exposure, Ca and Na increased (up to 0.6 g/L and 0.1 g/L, respectively), while phosphorus decreased (down to ~0.1 g/L) at all pH's (Figs 1C, 2C and S2C

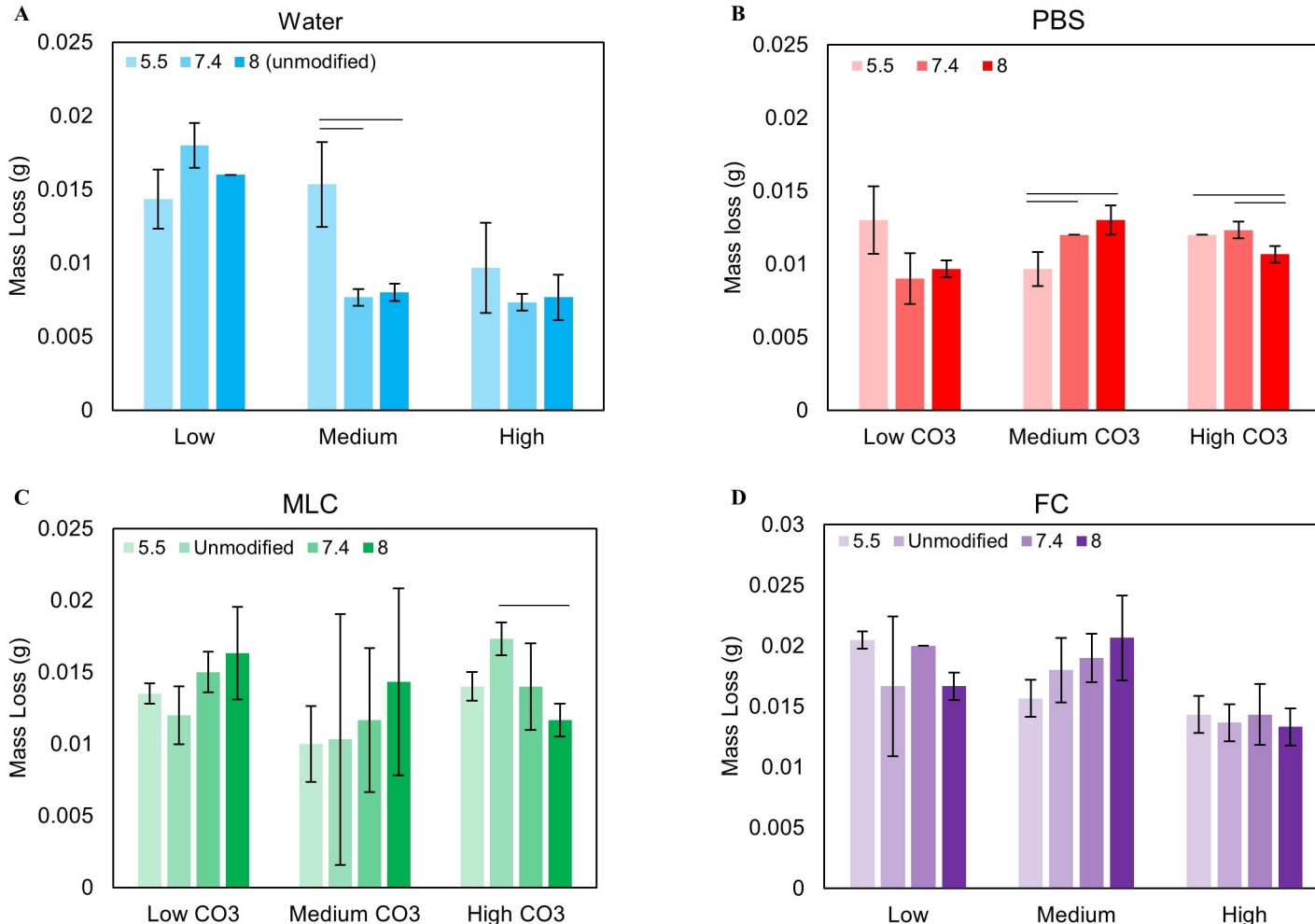

**Fig 4. Mass loss as a function of pH at each wt% carbonate for each solution.** Mass decreased for all solutions, indicating powder erosion **(A-D)**. Powders in FC had the greatest mass loss **(D)** while PBS and Water had the least **(A, B)**.

Fig). However, unlike PBS, there is no strong correlation between calcium, phosphorus, sodium levels, $CO_3^{2-}$ content, or any of the pH factors.

In all cases, the pH of the solutions increased with powder exposure. On average, pH 5.5, unmodified, pH 7.4 and 8.0 increased by 1, 0.4, 0.4, and 0.2 pH units, respectively (Fig 3C). The $pH_i$ is negatively correlated to $\Delta pH$.

The MLC caused an average mass loss of 0.013 g (Fig 4C), slightly larger than what was seen in water and PBS. This mass loss is generally unaffected by initial $CO_3^{2-}$ content or $pH_i$ (Fig 4C). The $CO_3^{2-}/PO_4^{3-}$ ratio generally decreased for the samples exposed to MLC solution as seen in PBS (Fig 5C). There are no clear trends relating initial $CO_3^{2-}$ levels in the powders to the amount of $CO_3^{2-}$ lost with exposure (Fig 5C). Contrarily, a decrease in $pH_i$ generally resulted in an increase in $CO_3^{2-}$ loss (Fig 5C).

In general, $\Delta d_C$ decreased and $\Delta d_A$ increased with exposure irrespectively of $pH_i$ and initial $CO_3^{2-}$ content. The MLC solution increased the crystal size by 0.03–1.56 nm and decreased $\varepsilon_{RMS}$ by 0.00182–0.014, except for pH 8 (S1C Fig and Fig 6C). Increases in final $CO_3^{2-}$ content and $pH_i$ were significantly correlated to a decrease in $d_A$ ($c_{CO3}$ = -0.772; p = 0.003 and $c_{pH}$ = -0.649; p = 0.02) (Table 1). There was a weak correlation between $d_C$ and final $CO_3^{2-}$ content

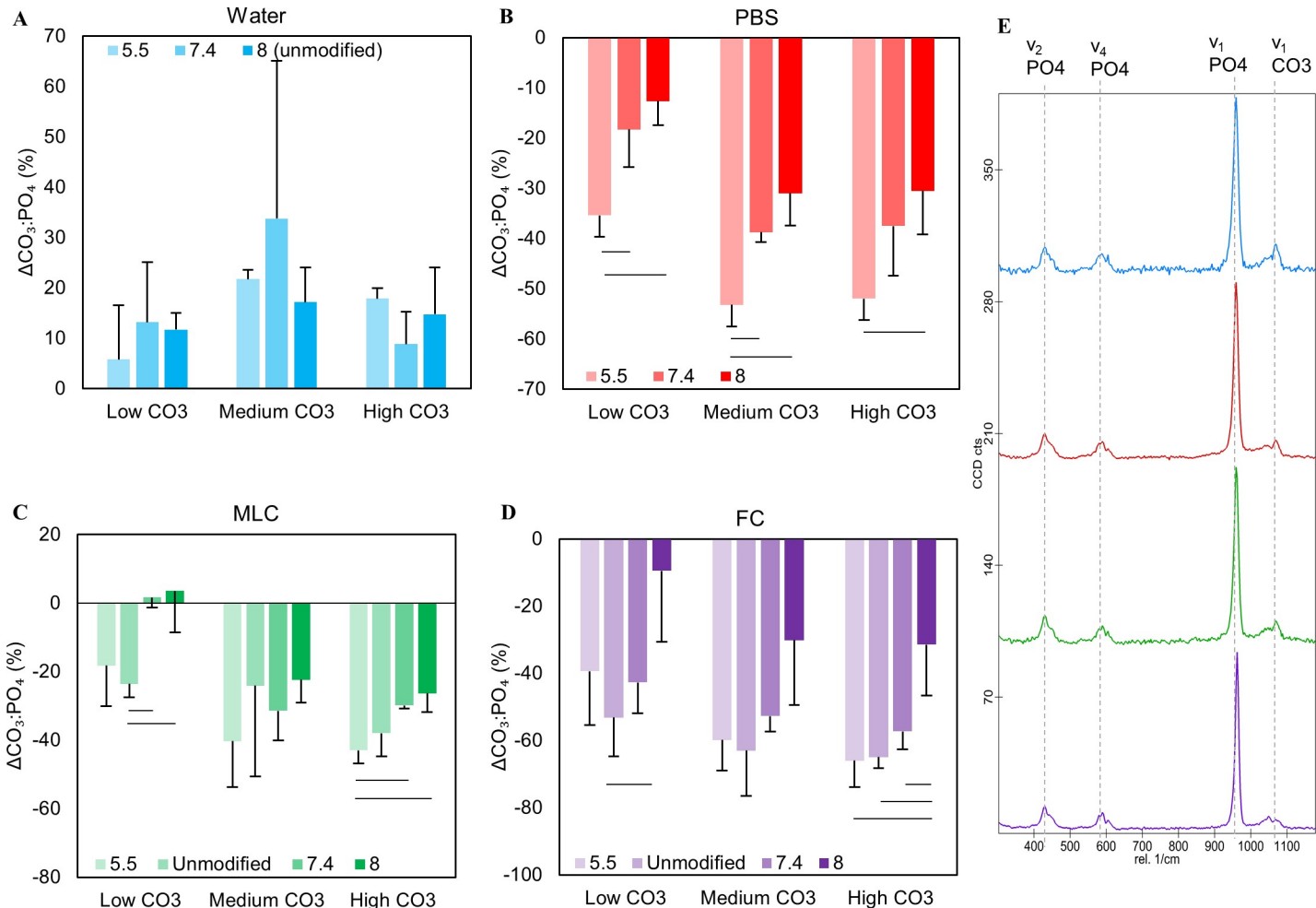

**Fig 5. Δ Carbonate: Phosphate ratios of CAP powders after exposure as a function of initial pH.** $CO_3^{2-}:PO_4^{2-}$ ratios decreased for PBS, MLC, and FC after solution exposure (**B-D**) while Water increased, suggesting an increase in carbonate amount (**A**). For PBS, $CO_3^{2-}:PO_4^{2-}$ significantly decreases as $pH_i$ decreases at all wt% $CO_3^{2-}$ (**B**). MLC and FC trended similarly to PBS (**C, D**), suggesting a loss of carbonate in apatite. Raman spectra of 6 wt% carbonated apatite after exposure in water (blue), PBS (red), MLC (green), and FC (purple) solutions at pH 5.5 (**E**).

and $pH_i$, (c = 0.750; p = 0.0049) (Table 1). The crystal size has a negative correlation with $pH_i$ (c = -0.612; p = 0.034) and a positive correlation with ΔpH (c = 0.615; p = 0.03) (Table 1). Conversely, $\varepsilon_{RMS}$ has a positive correlation with $pH_i$ (c = 0.574; p = 0.05) and a negative correlation with ΔpH (c = -0.655; p = 0.02) (Table 1).

## Effects of fluoride in solution

Before exposure, the FC solution had ~1.3 g/L of Na, ~0.84 g/L of K, ~0.62 g/L of P, and ~0.0033 g/L of Ca. This is the highest P of all solutions. P decreased to ~.23 g/L in the solution after powder exposure for all $CO_3^{2-}$ contents and $pH_i$ (Fig 2D). There is a strong correlation between the decrease in phosphorus in the solution as $pH_i$ decreased (c = 0.915; p = 0.000) while a decrease in phosphorus negatively correlated to $\Delta CO_3^{2-}/PO_4^{3-}$ (c = -0.556) (Table 1). Although there were no clear trends with $pH_i$ and $CO_3^{2-}$ content, there was a general increase in calcium in the solution after powder exposure (Fig 1D). Solution calcium content was negatively correlated with $pH_i$ (c = -0.531) (Table 1).

**Table 1. Correlation between all solution and powder conditions and outcomes.**

| | | pHi | Δcrystal size | Δstrain | 002 | 004 | 310 | ΔCa | ΔP | ΔNa | ΔK |
|---|---|---|---|---|---|---|---|---|---|---|---|
| **CO3/PO4f** | *PBS* | 0.463 | -0.427 | 0.446 | **0.828** | 0.792 | -0.579 | **-0.694** | -0.520 | 0.230 | -0.635 |
| | *MLC* | 0.415 | -0.289 | 0.117 | 0.038 | 0.750 | **-0.772** | 0.204 | 0.203 | 0.213 | **-0.604** |
| | FC | **0.664** | -0.347 | 0.207 | 0.484 | 0.743 | 0.246 | -0.074 | 0.490 | 0.069 | 0.281 |
| | *Water* | 0.118 | -0.042 | 0.214 | 0.500 | 0.388 | -0.516 | -0.495 | -0.459 | 0.555 | -0.497 |
| **% ΔCO3/PO4** | *PBS* | **-0.677** | -0.143 | 0.074 | 0.399 | 0.193 | 0.287 | 0.333 | -0.273 | 0.208 | 0.122 |
| | *MLC* | **-0.603** | 0.240 | -0.405 | 0.214 | 0.305 | -0.024 | -0.182 | -0.353 | 0.073 | -0.056 |
| | *FC* | **-0.707** | 0.361 | -0.468 | -0.173 | -0.500 | 0.063 | -0.085 | -0.556 | 0.305 | -0.044 |
| | *Water* | -0.251 | 0.010 | 0.051 | -0.413 | -0.213 | 0.259 | 0.321 | 0.301 | 0.024 | 0.500 |
| **pHi** | *PBS* | | -0.208 | 0.182 | 0.239 | 0.374 | **-0.663** | **-0.825** | 0.065 | 0.132 | -0.340 |
| | *MLC* | | **-0.612** | *0.574* | -0.020 | 0.199 | **-0.649** | 0.244 | 0.436 | 0.089 | -0.314 |
| | *FC* | | -0.217 | 0.188 | 0.450 | **0.674** | 0.145 | -0.531 | **0.915** | -0.067 | **0.631** |
| | *Water* | | -0.189 | 0.163 | 0.324 | 0.300 | 0.206 | -0.071 | 0.015 | -0.290 | -0.166 |
| **ΔpH** | *PBS* | | 0.058 | -0.079 | 0.203 | 0.037 | 0.460 | 0.597 | -0.326 | -0.033 | 0.106 |
| | *MLC* | | **0.615** | **-0.655** | -0.106 | 0.002 | 0.270 | -0.053 | -0.245 | 0.131 | 0.089 |
| | *FC* | | -0.531 | 0.104 | 0.156 | 0.458 | 0.517 | 0.082 | 0.279 | 0.364 | 0.279 |
| | *Water* | | 0.109 | 0.024 | -0.117 | -0.107 | -0.413 | -0.172 | -0.235 | 0.573 | -0.010 |
| **Δcrystal size** | *PBS* | | | **-0.967** | -0.276 | -0.309 | 0.425 | | | | |
| | *MLC* | | | **-0.942** | -0.420 | -0.404 | 0.317 | | | | |
| | *FC* | | | **-0.795** | -0.101 | -0.549 | -0.483 | | | | |
| | *Water* | | | **-0.932** | -0.220 | -0.418 | **-0.628** | | | | |
| **Δstrain** | *PBS* | | | | 0.192 | 0.307 | -0.507 | | | | |
| | *MLC* | | | | 0.337 | 0.303 | -0.169 | | | | |
| | *FC* | | | | 0.236 | **0.619** | 0.182 | | | | |
| | *Water* | | | | 0.044 | 0.342 | 0.491 | | | | |

Bolded is $p < 0.05$. Bolded and italicized is $p = 0.05$.

The pH of the FC solutions increased with powder exposure to significantly higher levels than in MLC and PBS (Fig 3D). On average, the FC solution increased by 0.9, 1.0, 1.1, and 0.8 pH units for pH 5.5, pH 7.4, 8.0, and unmodified, respectively. The ΔpH increased significantly with increased initial $CO_3^{2-}$ level and decreased $pH_i$ (Fig 3D).

The FC solution had an average mass loss of 0.016 g (Fig 4D); the largest mass loss of the 4 solutions studied. This mass loss did not vary with $pH_i$ or initial $CO_3^{2-}$ content, similar to MLC and PBS (Fig 4D).

In general, $CO_3^{2-}/PO_4^{3-}$ decreased at all $pH_i$ (Fig 5D) in FC, exhibiting the highest $CO_3^{2-}$ loss of all the solutions. No statistically significant relationship was identified between the change in $CO_3^{2-}/PO_4^{3-}$ and initial $CO_3^{2-}$ wt%. However, there was a significant negative correlation between $pH_i$ and percent change in $CO_3^{2-}/PO_4^{3-}$ post dissolution (c = -0.707; p = 0.01) (Table 1).

The $d_C$ and $d_A$ decreased with FC exposure for all conditions (Fig 7D). The $d_C$ decreased with decreasing $CO_3^{2-}/PO_4^{3-}$ ($c_{002}$ = 0.484; p = 0.0056) and $pH_i$ ($c_{002}$ = 0.450; p = 0.016) (Table 1). The $d_A$ was weakly positively correlated to the final $CO_3^{2-}/PO_4^{3-}$ (c = 0.246) and $pH_i$ (c = 0.145) (Table 1). In general, the crystal size increased up to 3.7 nm and $\varepsilon_{RMS}$ decreased down to -0.19 irrespective of $CO_3^{2-}$ content and $pH_i$ (S1D Fig and Fig 6D). Unlike other solutions, an increase in $\varepsilon_{RMS}$ is significantly correlated to the increase in $d_C$ in FC rinse ($c_{004}$ = 0.619; p = 0.032) (Table 1).

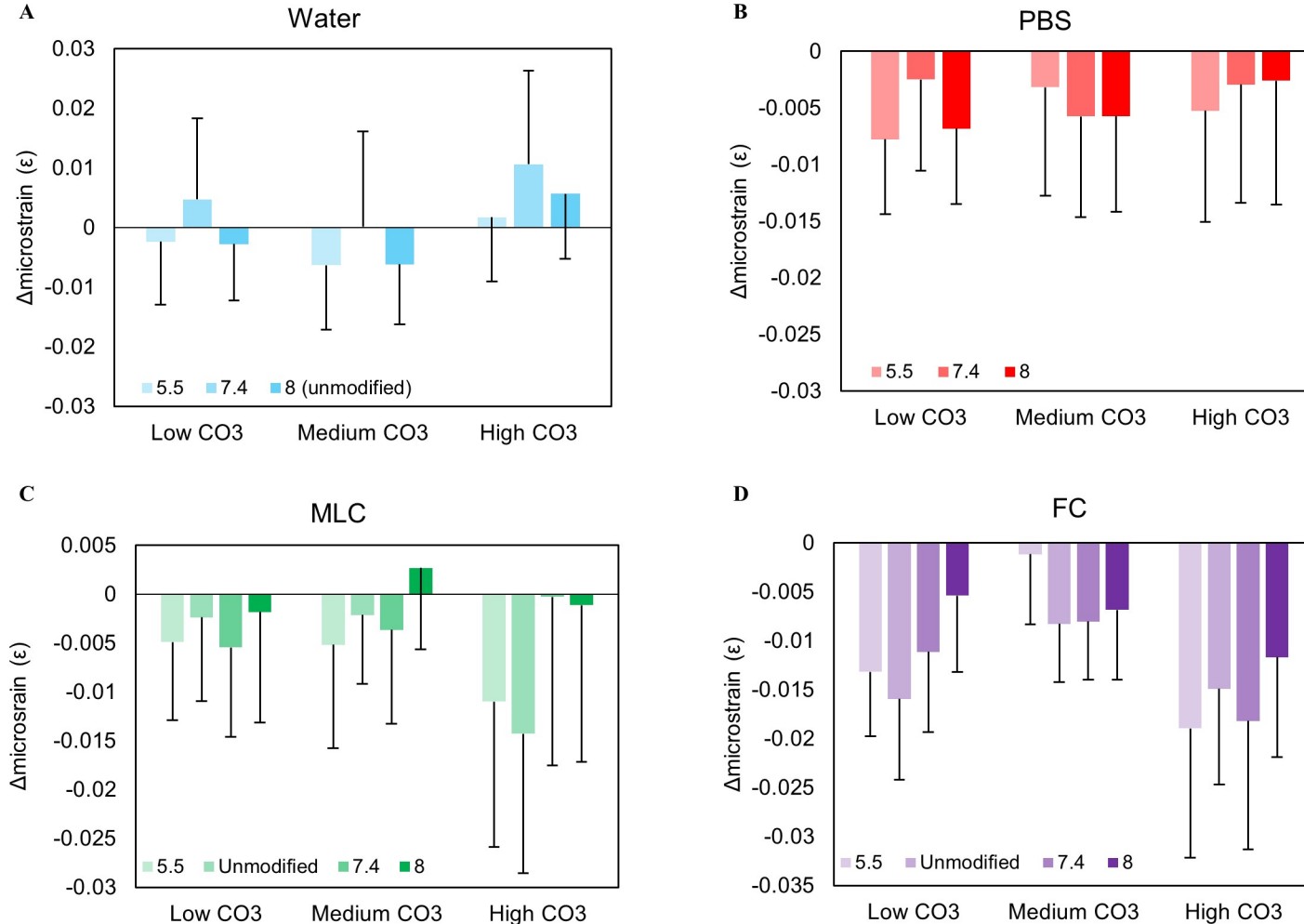

**Fig 6. The internal microstrain of CAP after solution exposure.** Small variations in microstrain were shown in water, suggesting no major change in structure (**A**). The microstrain decreased for PBS, MLC, and FC, although there are no trends relative to pH and wt% $CO_3^{2-}$ (**B-D**). This shows that the apatite is becoming a more perfect crystal after solution exposure. Error bars account for the standard deviations related to the y-intercept derived from the Halder-Wagner equation.

## Discussion

Ideally, xerostomia oral rinses would provide relief from the discomfort associated with dry mouth while maintaining tooth health. To address how the composition of xerostomia rinses impacts tooth health, numerous studies have examined the effect of different rinses and artificial salivas on tooth erosion [14–21, 27], which is defined as dissolution of mineralized tissue on the surface of teeth. Erosion tests usually involve placing dental tissues into the rinses of interest, followed by measurements of erosion via gravimetric analysis [20, 21], superficial hardness [18], microradiography [14, 16], fluorescence [17], or the amount of phosphorus dissolved [15]. These single outcome studies and variable measurement techniques have led to a contradictory body of literature where some have shown mineral loss while others show mineral gain [14–21]. Although these techniques are useful, these individual outcomes are not sufficient to understand the effects of xerostomia treatments on tooth health. In addition, the use of human or bovine teeth in most of these studies makes it difficult to identify precise dissolution mechanisms due to the pre-testing history of the tissues. To avoid these pitfalls, we have conducted a multi-technique analysis of the effect of the composition of clinically-relevant

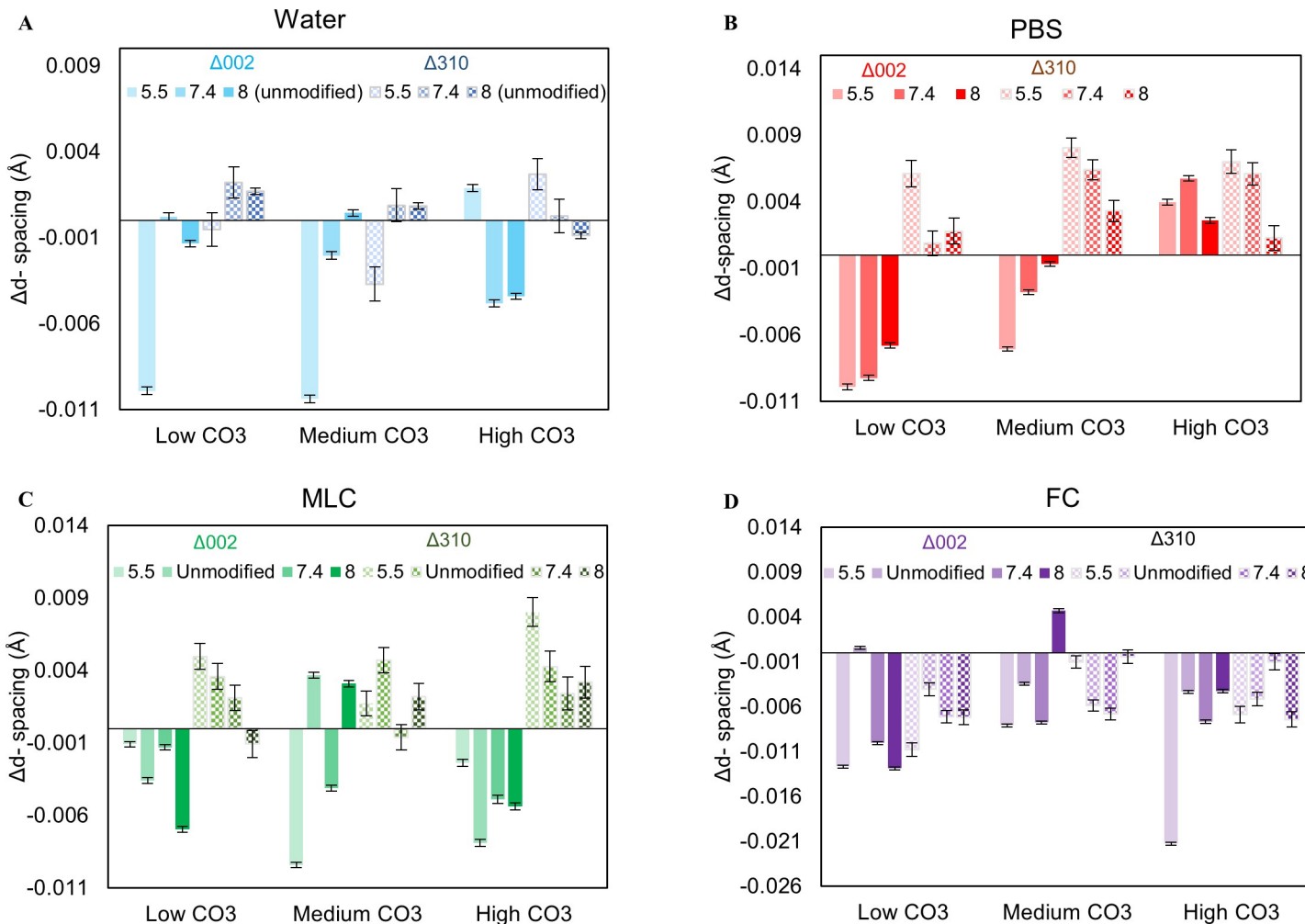

**Fig 7. D-spacing of the c-axis (002) and a-axis (004) of CAP after exposure.** Generally, the c-axis decreased, and the a-axis increased for PBS and MLC (**B, C**). For FC, both c-axis and a-axis decreased (**C**). These changes in d-spacing specify structural changes in apatite after solution exposure. Water exhibited variations of these axes, which is indicative of no structural differences of CAP (**A**). Error bars account for XRD peak fitting errors.

xerostomia rinses on well characterized biomimetic apatites. The goal is to determine how these treatments and the water, phosphorus content, presence of large mucin-like molecules, and fluoride in the rinses affect the dissolution, structure, and composition of the mineral.

## Water causes the formation of a carbonated surface layer

For many individuals who suffer from xerostomia, a first treatment option is rinsing or drinking water. However, rinsing with water provides only temporary relief for dry mouth, requiring constant swishing or drinking. To investigate the effect of water on dental mineral, we exposed biomimetic apatites to tap water. Water exposure led to an average 22% decrease in total sample mass. This large loss is likely due to dissolution of the crystals in the solution. Dental erosion does not occur at this rate in healthy individuals, suggesting that there are factors in the oral environment that were not accounted for here. These include increased surface area of the powders compared to natural tooth and a lack of protective pellicle formation *in vitro* [28, 10].

We found that pH had little effect on the extent of mass loss. The pH values selected here range from normal (pH 6–7), to more extreme salivary pHs (pH 5–8) during low and peak flow periods, respectively [10]. The lack of pH effects was unexpected, as it is generally accepted that solutions with lower pHs are more likely to dissolve mineral [20, 21]. However, other studies have similarly seen no effect of pH on mineral dissolution [15]. The mass loss was also independent of the initial carbonate content of the biomimetic apatite powders, despite its known effect on apatite solubility [29, 25].

This lack of pH effect may be related to the rapid ability of the powders to buffer the water solution. In all cases, the solution pH increased rapidly within the first hour of exposure to the apatites. Unlike mass loss, $\Delta$pH was dependent on both initial $pH_i$ and initial $CO_3^{2-}$ content of the powders, where $\Delta$pH increased as $pH_i$ decreased and $CO_3^{2-}$ content increased. These results suggest that the powders were able to (1) either release buffering moieties like bicarbonate, $HCO_3^-$, or (2) uptake protons to increase the solution pH as previously described both *in vitro* and *in vivo* [30–33]. The dissolution of the crystals may have released $HCO_3^-$, but it also released calcium and phosphorus into solution. The ratio of released calcium and phosphorus (Ca/P ratio) in solution has a value of ~1.8, indicating near stoichiometric dissolution as predicted by a Ca/P value of 1.67 for pure hydroxyapatite [34].

Despite this evidence of dissolution and buffering, Raman measurements of $CO_3^{2-}/PO_4^{3-}$ ratios in post-exposure crystals indicate that $CO_3^{2-}$ content increased in apatite with exposure. Therefore, the pH change must be explained either by buffering via $PO_4^{3-}$, as indicated by the increased P in the water after exposure, or by uptake of $H^+$ by the crystals [22]. The d-spacing, size, and $\varepsilon_{RMS}$ of the crystals after exposure showed no clear changes as indicated by XRD, suggesting that the additional $CO_3^{2-}$ seen in Raman is not being substituted into the crystal lattice. Thus, we hypothesize that the additional $CO_3^{2-}$ is maintained on the surface of the crystals as labile carbonate [34–36]. We suggest that $H^+$ ions from the solution may have been attracted to the negatively charged apatite, increasing the solution pH. The associated shift to a positive surface may have then recruited $CO_3^{2-}$ ions, possibly from dissolved atmospheric carbon dioxide, to the apatite surface creating a hydrated ion-rich surface layer with labile carbonate.

Although the tap water is fluoridated, the XRD results did not show the formation of fluorapatite. However, this may be expected as the water contained between 0.8 mg/L—1.2 mg/L of fluoride [37]. It is likely that the low fluoridation levels found in drinking water would not cause phase changes in the short time frame measured here. In conclusion, it appears that dental mineral when exposed to tap water acts to sequester $H^+$ ions, thus buffering the solution and forming a highly carbonated surface layer (Fig 8). The presence of this layer may serve to protect the mineral from future acid assaults via rapid release of $CO_3^{2-}$.

## PBS exposure causes the reprecipitation of less-carbonated apatites

PBS is not prescribed as an oral rinse in dental settings; however, it is the main storage solution in dental research. This phosphate buffered saline is believed to mitigate the dissolution of mineralized tissues due to its high phosphate content. However, post-exposure mass loss was greater in PBS than the water solution. The lower levels of calcium in PBS than in water may have contributed to the increased mass loss since a lack of calcium in saline solutions has been shown to cause calcium release from bone over time [38]. This is supported by the increase in calcium in the solution after powder exposure.

Similar to the water, all of the conditions exhibited an increase in pH with powder exposure. However, unlike water, the PBS-exposed apatite powders showed a decrease in $CO_3^{2-}$ content. This suggests that instead of sequestering $H^+$, the addition of P to the solution caused a release of $CO_3^{2-}$, most likely in the form of $HCO_3^-$, from the apatite into the solution to buffer

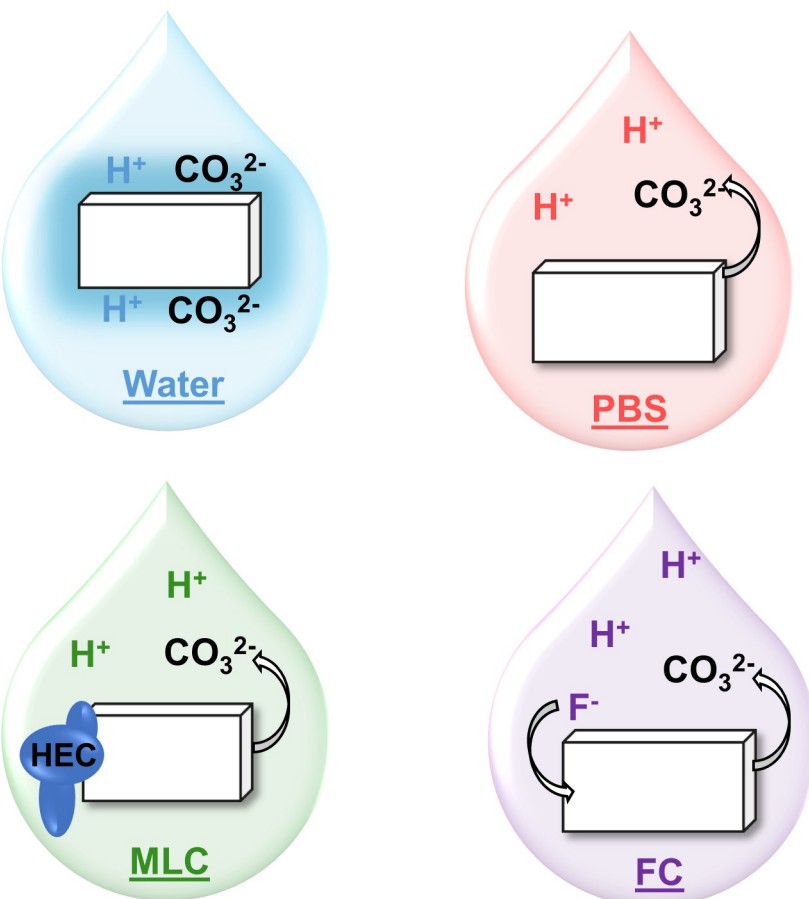

**Fig 8. Summary of dissolution mechanism of CAP in each solution.** Water does not affect apatite erosion and leaves labile carbonate on the crystal surface. The addition of phosphorus in solution allows carbonate to be released from apatite to buffer the acid in the solution. MLC has a similar mechanism, however, mucin-like molecules binding to apatite may affect apatite erosion. While carbonate is dissolved from apatite into FC to buffer the acid, fluoride may be incorporated into apatite during recrystallization.

the acid in PBS. This behavior has been previously reported in the literature for both teeth and bones [30, 31].

It is debated about whether the released $CO_3^{2-}$ is taken from the surface of the crystals as labile carbonate or from within the crystal lattice. XRD results showed that exposure to PBS increased the a-axis and decreased the c-axis lattice spacing of the apatite crystals. This points to a loss of B-type $CO_3^{2-}$, where carbonate is substituted for $PO_4^{3-}$, from the crystal lattice resulting in lattice deformation. Taken together, the data suggests that the apatite powders undergo dissolution, during which the $CO_3^{2-}$ is used for buffering, followed by reprecipitation of a less-carbonated apatite. Since B-type $CO_3^{2-}$ substitution requires carbonate to substitute for phosphate, reprecipitation of apatites with less carbonate would require additional phosphorus. This uptake of P by the recrystallized apatite is confirmed by a decrease in P ions in the PBS solution after exposure. XRD results also showed an increase in average crystal size and a decrease in $\varepsilon_{RMS}$ with exposure to PBS as expected for an apatite with decreased carbonate [39]. Overall, these results suggest that biomimetic apatites undergo dissolution followed by reprecipitation of less-carbonated crystals in PBS at all of the measured pHs (Fig 8). This change in the composition of the remaining apatite will significantly affect the solubility and mechanics of the tooth [39] resulting in stiffer and more corrosion resistant tissue.

## Large hydrophilic molecules in MLC may affect reprecipitation

MLC rinse (Biotène) is a commonly recommended oral rinse for xerostomia, with a recommended exposure of up to five 30-second rinses per day [40]. The MLC rinse is similar to PBS, with both containing an aqueous base with high phosphorus levels. However, MLC also contains large hydrophilic mucin-like molecules, like the hydroxyethyl cellulose (HEC), that are often added to provide a sense of hydration and lubrication [12]. Addition of mucins or mucin-like moieties has been shown to be protective against demineralization [41–44]. However, the apatite powders exposed to MLC rinse had a significant mass loss with exposure. While mucin-like molecules may have the ability to adsorb to apatite surfaces to possibly create a barrier [42], our Raman results suggest that HEC was only weakly adsorbed to the apatite surface since it was easily removed by three water rinses.

In terms of composition and structure, the powders exposed to MLC rinse showed general trends of decreasing $CO_3^{2-}$ content, $d_C$, $\varepsilon_{RMS}$, and solution phosphorus with exposure as seen in the PBS. However, the changes are smaller and the trends are less clear, indicating a modified dissolution and reprecipitation process. While mucins may limit demineralization, they may also limit remineralization, possibly due to complexation with calcium and increased viscosity preventing ion exchange [45–47]. This suggests that although the mucin-like moieties are not significantly affecting dissolution in this study, they may be affecting the reprecipitation process that may explain the highly variable mineral composition and structure in MLC exposed apatites. Taken together, these results show how mucin-like moieties may affect the delicate balance of demineralization and remineralization in the oral environment (Fig 8), and they should be considered as a potential mechanism of action affecting dental tissue composition.

## Fluoride in FC solution promoted reprecipitation of fluorapatite

The FC rinse (ACT) is another commonly recommended oral rinses for xerostomia with a suggested use of two 1-minute rinses everyday [48]. The composition of the FC rinse is relatively similar to the MLC rinse, with the addition of high levels of fluoride ([F-] 90 mg/L); approximately 75–112.5 times more concentrated than tap water. Historically, fluoride has been added to public sources of drinking water, toothpaste and other dental applications to promote the formation of fluorapatite, a fluoride-substituted apatite, in dental tissues [49]. The addition of fluoride to salivary rinses have shown to protect against demineralization and promote remineralization [17, 27, 50].

In this study, the apatite exposed to FC rinse behaved similarly to the apatites in PBS and MLC rinses in terms of mass loss, ΔpH, and mineral carbonate content. FC exhibited the largest mass loss of all solutions as well as the greatest increase in the final solution pH. The large decrease in $CO_3^{2-}$ content in the FC exposed powders suggests that these powders are undergoing classical dissolution-reprecipitation processes.

However, in this case the reprecipitated crystal is not a low-carbonate apatite, but fluorapatite, as suggested by the peak shift of the $v_1$ PO4 in the Raman spectra (Fig 5E). In addition, comparison of XRD spectra with standards from the PDF 2001 library showed that the post-exposure powders exhibit a greater resemblance to fluorapatite than carbonated apatite. The decrease in both $d_C$ and $d_A$ supports this removal of $CO_3^{2-}$ and formation of fluorapatite since fluorapatite exhibits a smaller $d_A$ than carbonated apatites [51]. This phase change decreased $\varepsilon_{RMS}$ and increased crystal size, as expected for the more crystalline fluorapatite. This reprecipitation also resulted in a decrease in solution phosphorus levels as it was likely taken up to form fluorapatite crystals (Fig 8). Since fluorapatite has a lower density than hydroxyapatite, this may explain the larger mass loss with FC rinse exposure. Overall, this highlights the need

to look beyond dissolution alone to understand how dental mineral will be affected by solution exposure.

The phase change of carbonated apatite to fluorapatite has significant dental benefits as fluorapatite exhibits decreased solubility and increased erosion resistance, therefore, protecting against acidogenic bacteria and reducing the risk of caries [49]. However, the addition of fluoride to the lattice structure also embrittles the mineral which can increase the risk of tissue fracture [52]. Thus, the balance between mechanics and erosion risk must be considered.

### Experimental limitations

Like any experimental *in vitro* study, there are limitations to the presented work. Regarding mass loss, the powders from all four solutions were scraped off the paper filters and weighed as such. Due to the small initial mass, it is possible that this led to non-significant data. Another limitation is that A-type $CO_3^{2-}$ substitutions are not accounted for in this study. Although small quantities of A-type $CO_3^{2-}$ are present in bones and teeth, B-type $CO_3^{2-}$ substitutions are predominantly found in these tissues [25, 53, 54], similar to our synthesized apatites. The changes in the lattice spacing from XRD indeed indicates that the bulk of the substitutions are B-type $CO_3^{2-}$ in our system and not A-type $CO_3^{2-}$. However, Raman is unable to distinguish between the two substitution types making it difficult to determine the role of A-type $CO_3^{2-}$ in this study. Finally, in attempting to isolate the effects of these varied xerostomia rinses, we must recognize that the composition of these solutions are complex, with more than one variable at play. While there are other ingredients in the FC and MLC rinses that could be affecting the dissolution seen in these experiments, the interactions between mucin-like molecules and fluoride on teeth have been previously shown to have the greatest effect on apatite mineral; therefore, we have chosen to focus on these factors instead of the many additional ingredients. We believe that this study plays a crucial role in beginning to elucidate properties of these components.

### Conclusion

The objective of this study was to elucidate the effects of clinically-relevant xerostomia rinse compositions on dentin-like biomimetic apatites. Four solutions were investigated to determine the effects of water, phosphorus content, mucin-like molecules, and fluoride on dental mineral. Biomimetic carbonated apatites with low, medium, and high levels of carbonate were exposed to each solution at pH 5.5, 7.4, 8.0, and unmodified for 72 hrs. In all cases, exposure to the solutions lead to mineral mass loss. There was no benefit to the addition of phosphorus, fluoride, or mucin-like molecules in terms of dissolution alone. However, the compositional and structural changes to the apatite were solution dependent (Fig 8). In the case of water, the structure of the crystals was minimally affected and instead, the apatites appeared to form a carbonate-rich surface layer. This was completely different from the three phosphorus containing solutions, in which there were significant decreases in carbonate content as well as changes to the lattice structure and crystal size. We propose that the addition of phosphorus shifts the mechanism to dissolution and reprecipitation. The addition of mucin-like molecules in the MLC rinse appeared to have a variable effect on the reprecipitation of the crystals, leading to unclear trends as compared to PBS. The addition of fluoride in the FC solution caused a phase change to fluorapatite during the reprecipitation. Taken together, this study shows that solution composition has a significant effect on the mode of dissolution and possible reprecipitation of dentin-like crystals. These changes to the crystals will have significant consequences on the minerals solubility and mechanics, affecting its response to future changes in pH and mechanical loading. This study highlights the need to look beyond dissolution when

examining the effects of dental treatments and will help clinicians to consider the complex consequences of the use of xerostomia oral rinses when advising patients.

## Supporting information

**S1 Fig. Change in crystallite size after solution exposure.** CAP has variations in crystal size after exposure to water, suggesting that the internal structure does not change (A). The crystallite size of CAP increased in PBS, MLC, and FC, indicating that CAP is becoming more crystalline after exposure (B-D). Error bars account for standard deviations related to the slope derived from the Halder-Wagner equation.
(TIF)

**S2 Fig. Change in sodium after exposure.** An increase in sodium in water may be due to a low sodium content in the pre-exposure water and indicates that CAP is releasing sodium during exposure (A). The amount of sodium in solution varied in PBS, MLC, and FC irrespective of $pH_i$ and powder carbonate content (B-D).
(TIF)

**S3 Fig. Change in potassium in the solution after exposure.** Potassium generally increased in PBS, revealing that apatite is releasing potassium (B). The opposite is shown for FC, where the decrease in potassium suggests that CAP is up taking into the structure during recrystallization to account for charge balance of the crystals (D). MLC and water were variable, implying that potassium has little effect in these solutions (A, C).
(TIF)

**S1 Table. List of ingredients for Biotene and ACT.**
(DOCX)

## Acknowledgments

We would like to acknowledge Anthony D'Angio for his analysis of the Raman standards. The Center for Environmental Sciences and Engineering at UConn provided the ICP-OES data. XRD data was acquired at the UConn Institute of Materials Science.

## Author Contributions

**Conceptualization:** Mikayla M. Moynahan, Alix C. Deymier.

**Data curation:** Mikayla M. Moynahan, Stephanie L. Wong.

**Formal analysis:** Mikayla M. Moynahan, Stephanie L. Wong.

**Funding acquisition:** Mikayla M. Moynahan, Alix C. Deymier.

**Investigation:** Mikayla M. Moynahan, Stephanie L. Wong.

**Methodology:** Stephanie L. Wong, Alix C. Deymier.

**Project administration:** Alix C. Deymier.

**Resources:** Alix C. Deymier.

**Supervision:** Alix C. Deymier.

**Validation:** Alix C. Deymier.

**Visualization:** Stephanie L. Wong.

**Writing – original draft:** Mikayla M. Moynahan, Stephanie L. Wong, Alix C. Deymier.

**Writing – review & editing:** Mikayla M. Moynahan, Stephanie L. Wong, Alix C. Deymier.

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
