## [Decision Letter · Decision Letter 0]

11 Feb 2021

PONE-D-20-33802

Beyond dissolution: xerostomia rinses affect composition and structure of biomimetic dental mineral

PLOS ONE

Dear Dr. Deymier,

Thank you for submitting your manuscript to PLOS ONE. After careful consideration, we feel that it has merit but does not fully meet PLOS ONE’s publication criteria as it currently stands. Therefore, we invite you to submit a revised version of the manuscript that addresses the points raised during the review process.

Specifically, there are a number of issues raised by the reviewer, affecting technical aspects of the work that need to be addressed before it can be furthered considered.

We look forward to receiving your revised manuscript.

Kind regards,

Oscar Millet

Academic Editor

PLOS ONE

Reviewers' comments:

Reviewer's Responses to Questions

**Comments to the Author**

1. Is the manuscript technically sound, and do the data support the conclusions?

Reviewer #1: Partly

2. Has the statistical analysis been performed appropriately and rigorously? 

Reviewer #1: Yes

3. Have the authors made all data underlying the findings in their manuscript fully available?

Reviewer #1: Yes

4. Is the manuscript presented in an intelligible fashion and written in standard English?

Reviewer #1: Yes

5. Review Comments to the Author

Reviewer #1: The objective of this study was to elucidate the effects of xerostomia rinse composition on dentin like biomimetic apatites.

The authors applied a multi-technique approach to understanding the effects of four different solutions containing: aqueous bases, high phosphate levels, mucin-like molecules, and fluoride. The combination of mass loss, pH, Raman spectroscopy, Inductively Coupled Plasma Optical Emission Spectrometry (ICP-OES), and X-ray diffraction (XRD) measurements allowed us to provide a comprehensive view of how these solutions modify dental tissues at a compositional and structural level.

It is an in vitro study and must appear in the title

The study design is not clear to readers. It is very narrative in some sections and should be synthesized

The authors use biotene and APC substances with different and many different active compounds are difficult to assess. Purified Water,Glycerin,Xylitol,Sorbitol, Propylene Glycol,Poloxamer ,sodium Benzoate…Hydroxyethyl Cellulose,Zingiber officinale (ginger) root extract,Angelica polymorpha sinensis root extract,Lactic acid,……………

Another limitation is that A-type CO3 substitutions are not accounted for in this study.

As strengths is the use of different techniques and some very interesting

Graphics have poor quality

I should provide some figure of the raman with the different components

6. PLOS authors have the option to publish the peer review history of their article (what does this mean?). If published, this will include your full peer review and any attached files.

Reviewer #1: No

---

## [Author Response · Author response to Decision Letter 0]

27 Mar 2021

Reviewer #1: 

Comment 1: The objective of this study was to elucidate the effects of xerostomia rinse composition on dentin like biomimetic apatites. The authors applied a multi-technique approach to understanding the effects of four different solutions containing: aqueous bases, high phosphate levels, mucin-like molecules, and fluoride. The combination of mass loss, pH, Raman spectroscopy, Inductively Coupled Plasma Optical Emission Spectrometry (ICP-OES), and X-ray diffraction (XRD) measurements allowed us to provide a comprehensive view of how these solutions modify dental tissues at a compositional and structural level.

Thank you for the supportive comments. We appreciate your understanding of the value of examining the effect of clinically relevant xerostomia treatments on the behavior of biomimetic apatites for dental health. 

Comment 1: It is an in vitro study and must appear in the title

We agree with the reviewer and have changed the title to read “Beyond dissolution: xerostomia rinses affect composition and structure of biomimetic dental mineral in vitro”.

Comment 2: The study design is not clear to readers. It is very narrative in some sections and should be synthesized.

We have made extensive revisions to the text in both organization and format to clarify the study design. The methods and discussion sections were rewritten to eliminate unnecessary verbiage and detail in order to reduce the prior narrative nature. Please see the manuscript with all tracked changed near the end of this document. 

Comment 3: The authors use biotene and APC substances with different and many different active compounds are difficult to assess. Purified Water,Glycerin,Xylitol,Sorbitol, Propylene Glycol,Poloxamer ,sodium Benzoate…Hydroxyethyl Cellulose,Zingiber officinale (ginger) root extract,Angelica polymorpha sinensis root extract,Lactic acid,……………

Thank you for your comment regarding the different ingredients. We agree with the reviewer that the additional components in the Biotene and ACT rinses complicate the exact mechanisms of dissolution/recrystallization of CHA in these two solutions. With that in mind, we wanted the solutions to be clinically relevant. This has been emphasized in the introduction on page 2-3, which now reads: 

“As a result, the current available research fails to truly elucidate the effect of oral rinse composition on dissolution and ignores the effect of these treatments on the remaining tooth mineral composition and structure. In addition, basic science studies examining the effects of specific ions/molecules on mineral content fail to provide clear clinically-relevant relationships about current treatment options.

In this study, we seek to elucidate the effect of clinically-relevant oral rinse composition on the dissolution, composition and structure of biomimetic dental apatites. We applied a multi-technique approach to understand the effects of four different clinically-relevant solutions containing: aqueous bases, high phosphate levels, mucin-like molecules, and fluoride.”

This is also emphasized in the methods sections on page 3-4 under the “Preparation of solutions” section: 

“To determine the effects of oral rinse composition on biomimetic dental mineral, four solutions were selected: (1) Tap water (aqueous base) (2) 1X phosphate buffered saline (PBS - aqueous base with additional phosphate), (3) Biotène Dry Mouth Oral Rinse® (mucin like containing - MLC), and (4) ACT® Anticavity Total Fluoride Mouthwash (fluoride containing - FC). The tap water was sourced from the Farmington, CT water supply. PBS was diluted from 10X PBS (Fisher Scientific, Waltham, MA) in MilliQ water. Biotène and ACT oral rinses were purchased from a local pharmacy. Biotène and ACT represent some of the most prescribed oral rinses for xerostomia. Although they have complex compositions, these 2 solutions were selected for the presence of mucin-like molecules such as hydroxyethyl cellulose in Biotène and the presence of high levels of fluoride in ACT (Table S1), making them unique and clinically-relevant solutions.”

We chose not to address the role of the many ingredients in Biotene and ACT that may play in dissolution/reprecipitation. This was in part due to low concentrations of the ingredients or previous literature data suggesting a lack of any effect. While we do not know the precise concentration of certain ingredients in the solutions due to patent rights, we do know the order of ingredients on the label, which are in the order of greatest concentration to lowest concentration [1]. The first 8 ingredients in Biotene are largely responsible for solution mouthfeel, taste, and solubility (Table S1). For example, poloxamer 407 is a surfactant [2], sodium benzoate is a preservative [2], xylitol/sorbitol are sugar-substitutes [3], and the viscosity of glycerin serves as an emulsifier and improves mouth feel [4,5]. There is limited or no information relating these compounds as well as others like parabens, root and flower extracts, and food coloring, to tooth dissolution and recrystallization. However, we do have some information about how some interact with tooth structures and dental health. 

The possible anti-cariogenic properties of xylitol and sorbitol have been extensively studied. It is suggested that these properties likely have to do with their interactions with oral bacteria; however, the literature results are mixed [3,6–12]. There is some evidence that xylitol may act as a Ca2+ transporter and affect mineralization [11]; however, the American Dental Association (ADA) currently classifies the xylitol data as insufficient [13]. Lactic acid in ACT might also be expected to have a demineralizing effect [14]. This might be in part responsible for the lower unmodified pH of the solution, although its position in the list of ingredients suggests a low concentration. Any acidification would be measured from the pH evaluations in this study. Food colorings, known as extrinsic chemogens, have been shown to accumulate in the surface deposits of teeth, specifically in the acquired pellicle layer [15,16]. We found that none of our powders exhibited a color change after exposure to the oral rinse solutions, likely due to a lack of pellicle layer in our in vitro powder model. While many of the ingredients in Biotene overlap with those in ACT, a notable exception is the addition of hydroxyethyl cellulose in Biotene. This mucin-like molecule has been implicated in the dissolution/remineralization mechanisms in previous studies as stated on page 17-18, making this a probable component of changing apatite properties. The role of hydroxyethyl cellulose in our studies remains unclear. Comparisons of Raman spectra of the apatites before and after exposure, as well as spectra from Biotene and ACT showed that there was generally no binding of additional components onto the apatites after rinsing. In a few cases however, Biotene exposed samples did exhibit Biotene associated peaks that match the spectra of hydroxyethyl cellulose [17] after a single rinse with Millipore water (Figure 1 ,figures are shown in the response to reviewers document at the end of this document). However, the Raman spectra of CHA showed no peaks from Biotene after a second rinse. The powders did not exhibit variations in crystal lattice suggesting the incorporation of large molecules, as shown by their XRD patterns (Figure 2). This suggests that there might be weak bonding between the hydroxyethyl cellulose and the apatite surfaces. This has led us to suggest on page 17-18 that interactions between this mucin-like molecule and the apatite in solution may serve as a protective coating during dissolution/recrystallization processes. 

In ACT, the first ingredient listed is 0.02% NaF (Table S1). Fluoride has long been known to substitute into apatite lattices to form fluorapatite, which exhibits decreased solubility and increased resistance to caries. We therefore propose that the biggest influencer of apatite dissolution/recrystallization process in ACT is the fluoride. Peaks shifts, decreases in peak width in our XRD data, as well as the significant carbonate loss from our Raman data indicate that there is a transition from the carbonated apatite to the more crystalline fluorapatite (Figure 2). This supports our theory that fluoride acts as a primary contributor to apatite dissolution/recrystallization behaviors. 

Based on the current available literature and our goal to elucidate the effects of the rinses as a whole on dissolution of dental mineral, we suggest that fluoride and MLCs are likely the most relevant components in the dissolution/recrystallization mechanisms of CHA in ACT and Biotene, respectively. This is concisely addressed in the discussion under “Experimental limitations” on page 19, which reads: “While there are other ingredients in the FC and MLC rinses that could be affecting the dissolution seen in these experiments, the interactions between mucin-like molecules and fluoride on teeth have been previously shown to have the greatest effect on apatite mineral; therefore, we have chosen to focus on these factors instead of the many additional ingredients. We believe that this study plays a crucial role in beginning to elucidate properties of these components.” 

Although it would be of great interest to examine the effects of each individual component, it is beyond the scope of this paper, whose goal was to elucidate how the rinses as a whole elicit clinically relevant dissolution responses on dental mineral. 

Comment 4: Another limitation is that A-type CO3 substitutions are not accounted for in this study.

We agree that it is important to understand how the carbonate is distributed across the crystals during dissolution and recrystallization, as changes in the amount and location of carbonate substitutions can significantly affect the structure, morphology, solubility, mechanics, and buffering ability of apatites [18–20]. Historically, B-type substitutions have been found to be the most prominent type of carbonate substitutions in biological apatites [21] due to its lower energy barrier for formation18. Conversely, A-type carbonate is more present in geologic and synthetic apatites exposed to high temperature and pressure conditions [22–24]. Since we create our apatites at 60℃ at local atmospheric pressure, this suggests that A-type carbonate is not the primary substitution involved in our teeth. However, there has been growing debate about the validity of methods like Fourier Transform Infrared Spectroscopy (FTIR), and Thermogravimetric Analysis (TGA) to identify differences in A- and B-type apatites. 

 Overlapping of FTIR peaks associated with B-, A- and labile (A2) carbonates in apatites can make it difficult to deconvolute individual peaks [25] (Figure 3). Our unpublished work with Dr. Claude Yoder examining bone apatites with FTIR has shown that relative amounts of A- vs B- type CO3 can vary depending on whether we deconvolve the v2 CO3 or the v3 CO3 peaks of the apatite. However, our XRD data, in agreement with other studies on biological apatites, indicate that our apatite models are more B-type apatites [26,27]. 

We are currently expanding our research interests to include the roles of A- and B- type carbonate on apatite dissolution/recrystallization and buffering using FTIR. We are also applying Nuclear Magnetic Resonance (NMR) to further differentiate A-type carbonate levels. Although extremely interesting, these additional techniques are beyond the scope of this work. We have identified the lack of information about carbonate location in the paper on page 19 which reads: 

“Another limitation is that A-type CO32- substitutions are not accounted for in this study. Although small quantities of A-type CO32- are present in bones and teeth, B-type CO32- substitutions are predominantly found in these tissues [25; 53; 54], similar to our synthesized apatites. The changes in the lattice spacing from XRD indeed indicates that the bulk of the substitutions are B-type CO32- in our system and not A-type CO32-. However, Raman is unable to distinguish between the two substitution types making it difficult to determine the role of A-type CO32- in this study.” 

Comment 5: Graphics have poor quality

Thank you for bringing this to our attention. The figures appeared blurry at submission but were of acceptable quality when the links were clicked. We have resubmitted the figures with even higher resolution here to hopefully address this issue. 

Comment 6: I should provide some figure of the raman with the different components

We appreciate this comment. We see the added value of including the Raman spectra of the apatites in the different solutions to our figures. We included the spectra of 6% carbonated apatite in all 4 solutions at pH 5.5 to Figure 5. The shift of the v1 PO4 peak at 960 cm-1 in the ACT sample as well as the narrowing of the peaks will further validate our findings regarding the fluoride contribution to CHA dissolution/recrystallization to a more fluorinated apatite. 

This has been referred to in the section titled “Fluoride in FC solution promoted reprecipitation of fluorapatite” on page 18, which reads: 

“However, in this case the reprecipitated crystal is not a low-carbonate apatite, but fluorapatite, as suggested by the peak shift of the v1 PO4 in the Raman spectra (Fig. 5E). In addition, comparison of XRD spectra with standards from the PDF 2001 library showed that the post-exposure powders exhibit a greater resemblance to fluorapatite than carbonated apatite. The decrease in both dC and dA supports this removal of CO32- and formation of fluorapatite since fluorapatite exhibits a smaller dA than carbonated apatites [49]. This phase change decreased εRMS and increased crystal size, as expected for the more crystalline fluorapatite. This reprecipitation also resulted in a decrease in solution phosphorus levels as it was likely taken up to form fluorapatite crystals (Fig. 8). Since fluorapatite has a lower density than hydroxyapatite, this may explain the larger mass loss with FC rinse exposure. Overall, this highlights the need to look beyond dissolution alone to understand how dental mineral will be affected by solution exposure.”

 

REFERENCES:

1. FDA. Why Are Food and Color Ingredients Added to Food? https://www.fda.gov/downloads/Food/FoodIngredientsPackaging/ucm094249.pdf 1–13 (2010).

2. Chemical, S. Mouthwash Ingredients. spec-001_ds_mouthwash_v3_new.pdf.

3. Maguire, A. & Rugg-Gunn, A. J. Xylitol and caries prevention-is it a magic bullet? BRITISH DENTAL JOURNAL VOLUME vol. 194 (2003).

4. Becker, L. C. et al. Expert Panel Safety Assessments Safety Assessment of Glycerin as Used in Cosmetics. doi:10.1177/1091581819883820.

5. Padmawar, A., Bhadoriya, U. & Padmawar, A. R. GLYCOL AND GLYCERIN: PIVOTAL ROLE IN HERBAL INDUSTRY AS SOLVENT/CO-SOLVENT. www.wjpmr.com (2018).

6. Riley, P., Moore, D., Ahmed, F., Sharif, M. O. & Worthington, H. V. Xylitol and Caries prevention. Cochrane Database of Systematic Reviews vol. 2015 (2015).

7. Gonçalves, N. C. L. A. V et al. Effect of xylitol:sorbitol on fluoride enamel demineralization reduction in situ. doi:10.1016/j.jdent.2005.12.008.

8. Cardoso, C. A. B. et al. Effect of xylitol varnishes on remineralization of artificial enamel caries lesions in vitro. doi:10.1016/j.jdent.2014.08.009.

9. Chunmuang, S., Jitpukdeebodintra, S., Chuenarrom, C. & Benjakul, P. Effect of xylitol and fluoride on enamel erosion in vitro. J. Oral Sci. 49, 293–297 (2007).

10. Rochel, I. D. et al. Effect of experimental xylitol and fluoride-containing dentifrices on enamel erosion with or without abrasion in vitro. J. Oral Sci. 53, 163–168 (2011).

11. Miake, Y., Saeki, Y., Takahashi, M. & Yanagisawa, T. Remineralization effects of xylitol on demineralized enamel. Journal of Electron Microscopy vol. 52 https://academic.oup.com/jmicro/article/52/5/471/1044146 (2003).

12. Tuncer, D., Onen, A. & Yazici, A. R. Effect of chewing gums with xylitol, sorbitol and xylitol-sorbitol on the remineralization and hardness of initial enamel lesions in situ. Dent. Res. J. (Isfahan). 11, 537–43 (2014).

13. New research shows clinical evidence unclear on effects of xylitol products preventing dental caries. https://www.ada.org/en/publications/ada-news/2015-archive/march/new-research-shows-clinical-evidence-unclear-on-effects-of-xylitol-products-preventing-dental-carie (2015).

14. Harper, R. A. et al. Acid-induced demineralisation of human enamel as a function of time and pH observed using X-ray and polarised light imaging. Acta Biomater. 120, 240–248 (2020).

15. Addy, M. & Moran, J. MECHANISMS OF STAIN FORMATION ON TEETH, IN PARTICULAR ASSOCIATED WITH METAL IONS AND ANTISEPTICS. Adv Dent Res vol. 9 (1995).

16. Watts, A. & Addy, M. Tooth discolouration and staining: A review of the literature. Br. Dent. J. 190, 309–316 (2001).

17. John Wiley & Sons, I. S. SpectraBase Compound. ID=J4sEjBlmZV SpectraBase.

18. Madupalli, H., Pavan, B. & Tecklenburg, M. M. J. Carbonate substitution in the mineral component of bone: Discriminating the structural changes, simultaneously imposed by carbonate in A and B sites of apatite. J. Solid State Chem. 255, 27–35 (2017).

19. Peccati, F., Bernocco, C., Ugliengo, P. & Corno, M. Properties and Reactivity toward Water of A Type Carbonated Apatite and Hydroxyapatite Surfaces. J. Phys. Chem. C 122, 3934–3944 (2018).

20. Yoder, C. H., Stepien, K. R. & Edner, T. M. A new model for the rationalization of the thermal behavior of carbonated apatites. J. Therm. Anal. Calorim. 140, 2179–2184 (2020).

21. Bigi, A. et al. Chemical and structural characterization of the mineral phase from cortical and trabecular bone. J. Inorg. Biochem. 68, 45–51 (1997).

22. Tacker, R. C. & Chris Tacker, R. Carbonate in igneous and metamorphic fluorapatite: Two type A and two type B substitutions. Am. Mineral. 93, 168–176 (2008).

23. Fleet, M. E. & Liu, X. Location of type B carbonate ion in type A-B carbonate apatite synthesized at high pressure. J. Solid State Chem. 177, 3174–3182 (2004).

24. Kubota, T., Nakamura, A., Toyoura, K. & Matsunaga, K. The effect of chemical potential on the thermodynamic stability of carbonate ions in hydroxyapatite. Acta Biomater. 10, 3716–3722 (2014).

25. Spizzirri, P. G., Cochrane, N. J., Prawer, S. & Reynolds, E. C. A Comparative Study of Carbonate Determination in Human Teeth Using Raman Spectroscopy. Caries Res. 46, 353–360 (2012).

26. Akkus, O., Adar, F. & Schaffler, M. B. Age-related changes in physicochemical properties of mineral crystals are related to impaired mechanical function of cortical bone. Bone 34, 443–453 (2004).

27. Deymier, A. C. et al. Protein-free formation of bone-like apatite: New insights into the key role of carbonation. Biomaterials 127, 75–88 (2017).

28. Aufort, J. et al. Atomic scale transformation of bone in controlled aqueous alteration experiments. Palaeogeogr. Palaeoclimatol. Palaeoecol. 526, 80–95 (2019).

---

## [Decision Letter · Decision Letter 1]

15 Apr 2021

Beyond dissolution: xerostomia rinses affect composition and structure of biomimetic dental mineral in vitro

PONE-D-20-33802R1

Dear Dr. Deymier,

We’re pleased to inform you that your manuscript has been judged scientifically suitable for publication and will be formally accepted for publication once it meets all outstanding technical requirements.

Kind regards,

Oscar Millet

Academic Editor

PLOS ONE

Additional Editor Comments (optional):

Reviewers' comments:

Reviewer's Responses to Questions

**Comments to the Author**

1. If the authors have adequately addressed your comments raised in a previous round of review and you feel that this manuscript is now acceptable for publication, you may indicate that here to bypass the “Comments to the Author” section, enter your conflict of interest statement in the “Confidential to Editor” section, and submit your "Accept" recommendation.

Reviewer #1: All comments have been addressed

2. Is the manuscript technically sound, and do the data support the conclusions?

Reviewer #1: Yes

3. Has the statistical analysis been performed appropriately and rigorously? 

Reviewer #1: Yes

4. Have the authors made all data underlying the findings in their manuscript fully available?

Reviewer #1: No

5. Is the manuscript presented in an intelligible fashion and written in standard English?

Reviewer #1: Yes

6. Review Comments to the Author

Reviewer #1: well done .

Changes have been made properly

Comment 1: It is an in vitro study and must appear in the title OK

The study design is not clear to readers. It is very narrative in some sections and should be synthesized . Has been added

The authors use biotene and APC substances with different and many different active compounds are difficult to assess. Has been added

Comment 4: Another limitation is that A-type CO3 substitutions are not accounted for in this study OK

Graphics have poor quality : Has been changed

7. PLOS authors have the option to publish the peer review history of their article (what does this mean?). If published, this will include your full peer review and any attached files.

Reviewer #1: No

---

## [Editor Report · Acceptance letter]

16 Apr 2021

PONE-D-20-33802R1 

Beyond dissolution: xerostomia rinses affect composition and structure of biomimetic dental mineral in vitro 

Dear Dr. Deymier:

I'm pleased to inform you that your manuscript has been deemed suitable for publication in PLOS ONE. Congratulations! Your manuscript is now with our production department. 

Kind regards, 

on behalf of

Dr. Oscar Millet 

Academic Editor

PLOS ONE